# LipSim: A Provably Robust Perceptual Similarity Metric

**Sara Ghazanfari**\*, **Alexandre Araujo**
**Prashanth Krishnamurthy**, **Farshad Khorrami**, **Siddharth Garg**
Department of Electronic and Computer Engineering
New York University
\*sg7457@nyu.edu

## Abstract

Recent years have seen growing interest in developing and applying perceptual similarity metrics. Research has shown the superiority of perceptual metrics over pixel-wise metrics in aligning with human perception and serving as a proxy for the human visual system. On the other hand, as perceptual metrics rely on neural networks, there is a growing concern regarding their resilience, given the established vulnerability of neural networks to adversarial attacks. It is indeed logical to infer that perceptual metrics may inherit both the strengths and shortcomings of neural networks. In this work, we demonstrate the vulnerability of state-of-the-art perceptual similarity metrics based on an ensemble of ViT-based feature extractors to adversarial attacks. We then propose a framework to train a robust perceptual similarity metric called **LipSim** (**Lip**schitz **Sim**ilarity Metric) with provable guarantees. By leveraging 1-Lipschitz neural networks as the backbone, LipSim provides guarded areas around each data point and certificates for all perturbations within an $\ell_2$ ball. Finally, a comprehensive set of experiments shows the performance of LipSim in terms of natural and certified scores and on the image retrieval application.

## 1 Introduction

Comparing data items and having a notion of similarity has long been a fundamental problem in computer science. For many years $\ell_p$ norms and other mathematically well-defined distance metrics have been used for comparing data items. However, these metrics fail to measure the semantic similarity between more complex data like images and are more focused on pixel-wise comparison. To address this problem perceptual distance metrics (Zhang et al., 2011; Fu et al., 2023) have been proposed that employ deep neural networks as a backbone to first compute embeddings, then apply traditional distance metrics on the embeddings of the data in the new space.

It is well-established that neural networks are susceptible to adversarial attacks (Goodfellow et al., 2014), That is, imperceptible variations of natural examples can be crafted to deliberately mislead models. Although perceptual metrics provide rich semantic interpretations compared to traditional metrics, they inherit the properties of neural networks and therefore their susceptibility to adversarial attacks (Kettunen et al., 2019; Sjögren et al., 2022; Ghildyal & Liu, 2022). Recent works have tried to address this problem by training robust perceptual metrics (Kettunen et al., 2019; Ghazanfari et al., 2023). However, these works rely on heuristic defenses and do not provide provable guarantees. Recent research has focused on designing and training neural networks with prescribed Lipschitz constants (Tsuzuku et al., 2018; Meunier et al., 2022; Wang & Manchester, 2023), aiming to improve and guarantee robustness against adversarial attacks. Promising techniques, like the SDP-based Lipschitz Layer (SLL) (Araujo et al., 2023), have emerged and allow to design of non-trivial yet efficient neural networks with pre-defined Lipschitz constant. Constraining the Lipschitz of neural has been known to induce properties such as stability in training (Miyato et al., 2018), robustness (Tsuzuku et al., 2018), and generalization (Bartlett et al., 2017).

Recently, the DreamSim metric (Fu et al., 2023) has been established as the state-of-the-art perceptual similarity metric. This metric consists of a concatenation of fine-tuned versions of ViT-

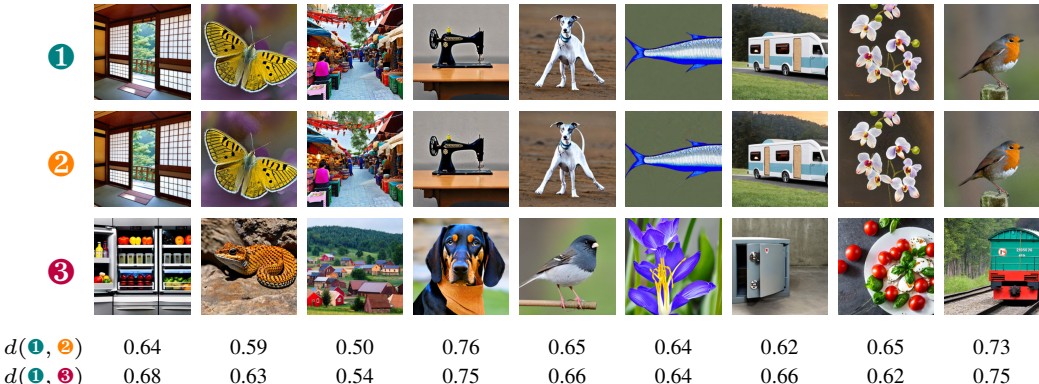

|   | | | | | | | | | |
|---|---|---|---|---|---|---|---|---|---|
| $d(\text{❶},\text{❷})$ | 0.64 | 0.59 | 0.50 | 0.76 | 0.65 | 0.64 | 0.62 | 0.65 | 0.73 |
| $d(\text{❶},\text{❸})$ | 0.68 | 0.63 | 0.54 | 0.75 | 0.66 | 0.64 | 0.66 | 0.62 | 0.75 |

Figure 1: **Demonstrating the effect of an attack on the alignment of DreamSim distance values with the real perceptual distance between images.** Instances of original and adversarial reference images from the NIGHT dataset are shown in the first and second rows and the DreamSim distance between them (*i.e.*, $d(\text{❶},\text{❷})$) is reported below. To get a sense of how big the distance values are, images that have the same distance from the original images are shown in the third row (*i.e.*, $d(\text{❶},\text{❸}) = d(\text{❶},\text{❷})$). Obviously, the first and third rows include semantically different images, whereas the images on the first and second rows are perceptually identical.

based embeddings, namely, DINO (Caron et al., 2021), CLIP (Radford et al., 2021), and Open CLIP (Cherti et al., 2023). To compute the distance between two images, DreamSim measures the cosine similarity distance between these ViT-based embeddings.

In this work, we initially demonstrate with a series of experiments that the DreamSim metric is not robust to adversarial examples. Consequently, it could be easy for an attacker to bypass important filtering schemes based on perceptual hash, copy detection, etc. Then, to tackle this problem, we propose LipSim, the first perceptual similarity metric *with provable guarantees*. Building on the DreamSim metric and recent advances in 1-Lipschitz neural networks, we propose a novel student-teacher approach with a Lipschitz-constrained student model. Specifically, we train a 1-Lipschitz feature extractor (student network) based on the state-of-the-art SLL architecture. The student network is trained to mimic the outputs of the embedding of the DreamSim metric, thus distilling the intricate knowledge captured by DreamSim into the 1-Lipschitz student model. After training the 1-Lipschitz feature extractor on the ImageNet-1k dataset, we fine-tune it on the NIGHT dataset which is a two-alternative forced choice (2AFC) dataset that seeks to encode human perceptions of image similarity (more explanation and some instances of NIGHT dataset are presented in Appendix B). By combining the capabilities of DreamSim with the provable guarantees of a Lipschitz network, our approach paves the way for a certifiably robust perceptual similarity metric. Finally, we demonstrate good natural accuracy and state-of-the-art certified robustness on the NIGHT dataset. Our contributions can be summarized as follows:

- We investigate the vulnerabilities of state-of-the-art ViT-based perceptual distance including DINO, CLIP, OpenCLIP, and DreamSim Ensemble. The vulnerabilities are highlighted using APGD (Croce & Hein, 2020) on the 2AFC score which is an index for human alignment and PGD attack against the distance metric and calculating the distance between an original image and its perturbed version.

- We propose a framework to train the first certifiably robust distance metric, **LipSim**, which leverages a pipeline composed of 1-Lipschitz feature extractor, projection to the unit $\ell_2$ ball and cosine distance to provide certified bounds for the perturbations applied on the reference image.

- We show by a comprehensive set of experiments that not only LipSim provides certified accuracy for a specified perturbation budget, but also demonstrates good performance in terms of natural 2AFC score and accuracy on image retrieval which is a serious application for perceptual metrics.

## 2 RELATED WORKS

**Similarity Metrics.** Low-level metrics including $\ell_p$ norms, PSNR as point-wise metrics, SSIM (Wang et al., 2004) and FSIM (Zhang et al., 2011) as patch-wise metrics fail to capture the high-level structure and the semantic concept of more complicated data points like images. In order to overcome this challenge the perceptual distance metrics were proposed. In the context of perceptual distance metrics, neural networks are used as feature extractors, and the low-level metrics are employed in the embeddings of images in the new space. The feature extractors used in recent work include a convolutional neural network as proposed by Zhang et al. (2018) for the LPIPS metric, or an ensemble of ViT-based models (Radford et al., 2021; Caron et al., 2021) as proposed by Fu et al. (2023) for DreamSim. As shown by experiments the perceptual similarity metrics have better alignment with human perception and are considered a good proxy for human vision.

**Adversarial Attacks & Defenses.** Initially demonstrated by Szegedy et al. (2013), neural networks are vulnerable to adversarial attacks, *i.e.*, carefully crafted small perturbations that can fool the model into predicting wrong answers. Since then a large body of research has been focused on generating stronger attacks (Goodfellow et al., 2014; Kurakin et al., 2018; Carlini & Wagner, 2017; Croce & Hein, 2020; 2021) and providing more robust defenses (Goodfellow et al., 2014; Madry et al., 2017; Pinot et al., 2019; Araujo et al., 2020; 2021; Meunier et al., 2022). To break this pattern, certified adversarial robustness methods were proposed. By providing mathematical guarantees, the model is theoretically robust against the worst-case attack for perturbations smaller than a specific perturbation budget. Certified defense methods fall into two categories. Randomized Smoothing (Cohen et al., 2019; Salman et al., 2019) turns an arbitrary classifier into a smoother classifier, then based on the Neyman-Pearson lemma, the smooth classifier obtains some theoretical robustness against a specific $\ell_p$ norm. Despite the impressive results achieved by randomized smoothing in terms of natural and certified accuracy (Carlini et al., 2023), the high computational cost of inference and the probabilistic nature of the certificate make it difficult to deploy in real-time applications. Another direction of research has been to leverage the Lipschitz property of neural networks (Hein & Andriushchenko, 2017; Tsuzuku et al., 2018) to better control the stability and robustness of the model. Tsuzuku et al. (2018) highlighted the connection between the certified radius of the network with its Lipschitz constant and margin. As calculating the Lipschitz constant of a neural network is computationally expensive, a body of work has focused on designing 1-Lipschitz networks by constraining each layer with its spectral norm (Miyato et al., 2018; Farnia et al., 2018), replacing the normalized weight matrix by an orthogonal ones (Li et al., 2019; Prach & Lampert, 2022) or designing 1-Lipschitz layer from dynamical systems (Meunier et al., 2022) or control theory arguments (Araujo et al., 2023; Wang & Manchester, 2023).

**Vulnerabilities and Robustness of Perceptual Metrics.** Investigating the vulnerabilities of perceptual metrics has been overlooked for years since the first perceptual metric was proposed. As shown in (Kettunen et al., 2019; Ghazanfari et al., 2023; Sjögren et al., 2022; Ghildyal & Liu, 2022) perceptual similarity metrics (LPIPS (Zhang et al., 2018)) are vulnerable to adversarial attacks. (Sjögren et al., 2022) presents a qualitative analysis of deep perceptual similarity metrics resilience to image distortions including color inversion, translation, rotation, and color stain. Finally (Luo et al., 2022) proposes a new way to generate attacks to similarity metrics by reducing the similarity between the adversarial example and its original while increasing the similarity between the adversarial example and its most dissimilar one in the minibatch. To introduce robust perceptual metrics, (Kettunen et al., 2019) proposes e-lpips which uses an ensemble of random transformations of the input image and demonstrates the empirical robustness using qualitative experiments. (Ghildyal & Liu, 2022) employs some modules including anti-aliasing filters to provide robustness to the vulnerability of LPIPS to a one-pixel shift. More recently (Ghazanfari et al., 2023) proposes R-LPIPS which is a robust perceptual metric achieved by adversarial training Madry et al. (2017) over LPIPS and evaluates R-LPIPS using extensive qualitative and quantitative experiments on BAPPS (Zhang et al., 2018) dataset. Besides the aforementioned methods that show empirical robustness, (Kumar & Goldstein, 2021; Shao et al., 2023) propose methods to achieve certified robustness on perceptual metrics based on randomized smoothing. For example, Kumar & Goldstein (2021) proposed center smoothing which is an approach that provides certified robustness for structure outputs. More precisely, the center of the ball enclosing at least half of the perturbed points in output space is considered as the output of the smoothed function and is proved to be robust to input perturbations

bounded by an $\ell_2$-size budget. The proof requires the distance metric to hold symmetry property and triangle inequality. As perceptual metrics generally don't hold the triangle inequality, the triangle inequality approximation is used which makes the bound excessively loose. In Shao et al. (2023), the same enclosing ball is employed however, the problem is mapped to a binary classification setting to leverage the certified bound as in the randomized smoothing paper (by assigning one to the points that are in the enclosing ball and zero otherwise). Besides their loose bound, these methods are computationally very expensive due to the Monte Carlo sampling for each data point.

## 3 BACKGROUND

**Lipschitz Networks.** After the discovery of the vulnerability of neural networks to adversarial attacks, one major direction of research has focused on improving the robustness of neural networks to small input perturbations by leveraging Lipschitz continuity. This goal can be mathematically achieved by using a Lipschitz function. Let $f$ be a Lipschitz function with $L_f$ Lipschitz constant in terms of $\ell_2$ norm, then we can bound the output of the function by $\|f(x) - f(x + \delta)\|_2 \leq L_f \|\delta\|_2$. To achieve stability using the Lipschitz property, different approaches have been taken. One efficient way is to design a network with 1-Lipschitz layers which leads to a 1-Lipschitz network (Meunier et al., 2022; Araujo et al., 2023; Wang & Manchester, 2023).

**State of the Art Perceptual Similarity Metric.** DreamSim is a recently proposed perceptual distance metric (Fu et al., 2023) that employs cosine distance on the concatenation of feature vectors generated by an ensemble of ViT-based representation learning methods. More precisely DreamSim is a concatenation of embeddings generated by DINO (Caron et al., 2021), CLIP (Radford et al., 2021), and Open CLIP (Cherti et al., 2023). Let $f$ be the feature extractor function, the DreamSim distance metric $d(x_1, x_2)$ is defined as:

$$d(x_1, x_2) = 1 - S_c(f(x_1), f(x_2)) \tag{1}$$

where $S_c(x_1, x_2)$ is the cosine similarity metric. To fine-tune the DreamSim distance metric, the NIGHT dataset is used which provides two variations, $x_0$ and $x_1$ for a reference image $x$, and a label $y$ that is based on human judgments about which variation is more similar to the reference image $x$ (some instances and more explanation of the NIGHT dataset are deferred to Appendix B). Supplemented with this dataset, the authors of DreamSim turn the setting into a binary classification problem. More concretely, given a triplet $(x, x_0, x_1)$, and a feature extractor $f$, they define the following classifier:

$$h(x) = \begin{cases} 1, & d(x, x_1) \leq d(x, x_0) \\ 0, & d(x, x_1) > d(x, x_0) \end{cases} \tag{2}$$

Finally, to better align DreamSim with human judgment, given the triplet $(x, x_0, x_1)$, they optimize a hinge loss based on the difference between the perceptual distances $d(x, x_1)$ and $d(x, x_1)$ with a margin parameter. Note that the classifier $h$ has a dependency on $f$, $d$ and each input $x$ comes as triplet $(x, x_0, x_1)$ but to simplify the notation we omit all these dependencies.

## 4 LIPSIM: LIPSCHITZ SIMILARITY METRIC

In this section, we present the theoretical guarantees of LipSim along with the technical details of LipSim architecture and training.

### 4.1 A PERCEPTUAL METRIC WITH THEORETICAL GUARANTEES

**General Robustness for Perceptual Metric.** A perceptual similarity metric can have a lot of important use cases, *e.g.*, image retrieval, copy detection, etc. In order to make a robust perceptual metric we need to ensure that when a small perturbation is added to the input image, the output distance should not change a lot. In the following, we demonstrate a general robustness property when the feature extractor is 1-Lipschitz and the embeddings lie on the unit $\ell_2$ ball, *i.e.*, $\|f(x)\|_2 = 1$.

**Proposition 1.** *Let $f : \mathcal{X} \to \mathbb{R}^k$ be a 1-Lipschitz feature extractor and $\|f(x)\|_2 = 1$, let $d$ be a distance metric defined as in Equation 1 and let $\delta \in \mathcal{X}$ and $\varepsilon \in \mathbb{R}^+$ such that $\|\delta\|_2 \leq \varepsilon$. Then, we have,*

$$|d(x_1, x_2) - d(x_1 + \delta, x_2)| \leq \|\delta\|_2 \tag{3}$$

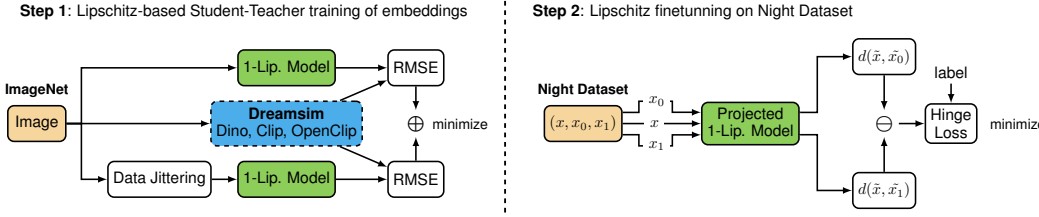

Figure 2: Two-step process for training the LipSim perceptual similarity metric. First (left), a distillation on the ImageNet dataset is performed where DreamSim acts as the teacher model, and a 1-Lipschitz neural network (*i.e.*, the feature extractor) is learned to mimic DreamSim embeddings. To reduce color bias with the Lipschitz network, we use two different dataset augmentation schemes: a simple random flip and a jittered data augmentation technique that varies the brightness, contrast, hue, and saturation of the image. Second (right), the 1-Lipschitz neural network with projection is then fine-tuned on the NIGHT dataset with a hinge loss.

The proof is deferred to Appendix A. This proposition implies that when the feature extractor is 1-Lipschitz and its output is projected on the unit $\ell_2$ ball then the composition of the distance metric $d$ and the feature extractor, *i.e.*, $d \circ f$, is also 1-Lipschitz with respect to its first argument. This result provides some general stability results and guarantees that the distance metric cannot change more than the norm of the perturbation.

**Certified Robustness for 2AFC datasets.** We aim to go even further and provide certified robustness for perceptual similarity metrics with 2AFC datasets, *i.e.*, in a classification setting. In the following, we show that with the same assumptions as in Proposition 1, the classifier $h$ can obtain *certified* accuracy. First, let us define a soft classifier $H : \mathcal{X} \to \mathbb{R}^2$ with respect to some feature extractor $f$ as follows:

$$H(x) = [d(x, x_1), d(x, x_0)] \tag{4}$$

It is clear that $h(x) = \arg\max_{i \in \{0,1\}} H_i(x)$ where $H_i$ represent the $i$-th value of the output of $H$. The classifier $h$ is said to be certifiably robust at radius $\epsilon \geq 0$ at point $x$ if for all $\|\delta\|_2 \leq \epsilon$ we have:

$$h(x + \delta) = h(x) \tag{5}$$

Equivalently, one can look at the margin of the soft classifier: $M_{H,x} := H_y(x) - H_{1-y}(x)$ and provide a provable guarantee that:

$$M_{H,x+\delta} > 0 \tag{6}$$

**Theorem 1** (Certified Accuracy for Perceptual Distance Metric). *Let $H : \mathcal{X} \to \mathbb{R}^2$ be the soft classifier as defined in Equation 4. Let $\delta \in \mathcal{X}$ and $\varepsilon \in \mathbb{R}^+$ such that $\|\delta\|_2 \leq \varepsilon$. Assume that the feature extractor $f : \mathcal{X} \to \mathbb{R}^k$ is 1-Lipschitz and that for all $x$, $\|f(x)\|_2 = 1$, then we have the following result:*

$$M_{H,x} \geq \varepsilon \|f(x_0) - f(x_1)\|_2 \implies M_{H,x+\delta} \geq 0 \tag{7}$$

The proof is deferred to Appendix A. Based on Theorem 1, and assuming $x_1 \neq x_0$, the certified radius for the classier $h$ at point $x$ can be computed as follows:

$$R(h, x) = \frac{M_{H,x}}{\|f(x_0) - f(x_1)\|_2} \tag{8}$$

Theorem 1 provides the necessary condition for a provable perceptual distance metric without changing the underlying distance on the embeddings (*i.e.*, cosine similarity). This result has two key advantages. First, as in Tsuzuku et al. (2018), computing the certificate at each point only requires efficient computation of the classifier margin $H$. Leveraging Lipschitz continuity enables efficient certificate computation, unlike the randomized smoothing approach of Kumar & Goldstein (2021) which requires Monte Carlo sampling for each point. Second, the bound obtained on the margin to guarantee the robustness is in fact tighter than the one provided by Tsuzuku et al. (2018). Recall Tsuzuku et al. (2018) result states that for a L-Lipschitz classifier $H$, we have:

$$M_{H,x} \geq \varepsilon \sqrt{2} L \implies M_{H,x+\delta} \geq 0 \tag{9}$$

Given that the Lipschitz constant of $H^1$ is $\sqrt{2}$, this lead to the following bound:

$$M_{H,x} \geq 2\varepsilon \geq \varepsilon \|f(x_0) - f(x_1)\|_2 \tag{10}$$

simply based on the triangle inequality and the assumption that $\|f(x)\|_2 = 1$.

## 4.2 LipSim Architecture & Training

To design a reliable feature extractor that can be used with Proposition 1 and Theorem 1, we combined a 1-Lipschitz neural network architecture with an Euclidean projection. Let $f : \mathcal{X} \to \mathbb{R}^k$ such that:

$$f(x) = \pi_{B_2(0,1)} \circ \beta \circ \phi^{(l)} \circ \cdots \circ \phi^{(1)}(x) \tag{11}$$

where $l$ is the number of layers, $\pi_{B_2(0,1)}$ is a projection on the unit $\ell_2$ ball, *i.e.*, $\pi_{B_2(0,1)}(x) = \arg\min_{z \in B_2(0,1)} \|x - z\|_2$ and where the layers $\phi$ are the SDP-based Lipschitz Layers (SLL) proposed by Araujo et al. (2023):

$$\phi(x) = x - 2W \operatorname{diag}\left(\sum_{j=1}^n |W^\top W|_{ij}\right)^{-1} \sigma(W^\top x + b), \tag{12}$$

where $W$ is a parameter matrix being either dense or a convolution, $\{q_i\}$ forms a diagonal scaling matrix and $\sigma$ is the ReLU nonlinear activation.

To apply Theorem 1 for certification, we need $\|f(x)\|_2 = 1$ with $f$ 1-Lipschitz. To respect these conditions, we need $\left\|\beta \circ \phi^{(l)} \circ \cdots \circ \phi^{(1)}(x)\right\|_2 \geq 1$ in order for the projection $\pi_{B_2(0,1)}$ to be 1-Lipschitz. The mapping $\beta : x \mapsto x + b$ is a 1-Lipschitz translation which is set to increase the norm of the embedding (*i.e.*, ideally above one) before the Euclidean projection. However, we do not have a guarantee that the norm of the embedding after the $\beta$ projection is above one, and therefore concerning the assumption of Theorem 1, in such cases, we abstain from the prediction. In practice, however, for all the samples of the NIGHT dataset $\left\|\beta \circ \phi^{(l)} \circ \cdots \circ \phi^{(1)}(x)\right\|_2 \geq 1$ and thus the abstain percentage is equal to zero.

**Proposition 2.** *The neural network $f : \mathcal{X} \to \mathbb{R}^k$ describe in Equation 11 is 1-Lipschitz and for all $x$ with $\|f(x)\|_2 = 1$ under the assumption that $\left\|\beta \circ \phi^{(l)} \circ \cdots \circ \phi^{(1)}(x)\right\|_2 \geq 1$*

*Proof of Proposition 2.* The proof is a straightforward application of Theorem 3 of Araujo et al. (2023) and Corollary 2.2.3 of Nesterov et al. (2018). □

**Two-step Process for Training LipSim.** LipSim aims to provide good image embeddings that are less sensitive to adversarial perturbations. We train LipSim in two steps, similar to the DreamSim approach. Recall that DreamSim first concatenates the embeddings of three ViT-based models and then fine-tunes the result on the NIGHT dataset. However, to obtain theoretical guarantees, we cannot use the embeddings of three ViT-based models because they are not generated by a 1-Lipschitz feature extractor. To address this issue and avoid self-supervised schemes for training the feature extractor, we leverage a distillation scheme on the ImageNet dataset, where DreamSim acts as the teacher model and we use a 1-Lipschitz neural network (without the $\ell_2$ unit ball projection) as a student model. This first step is described on the left of Figure 2. In the second step, we fine-tuned the 1-Lipschitz neural network with projection on the NIGHT dataset using a hinge loss to increase margins and therefore robustness, as in Araujo et al. (2023). This second step is described on the right of Figure 2.

## 5 Experiments

In this section, we present a comprehensive set of experiments to first highlight the vulnerabilities of DreamSim which is the state-of-the-art perceptual distance metric, and second to demonstrate the certified and empirical robustness of LipSim to adversarial attacks.

---

[1] Recall the Lipschitz of the concatenation. Let $f$ and $g$ be $L_f$ and $L_g$-Lipschitz, then the function $x \mapsto [f(x), g(x)]$ can be upper bounded by $\sqrt{L_f^2 + L_g^2}$-Lipschitz

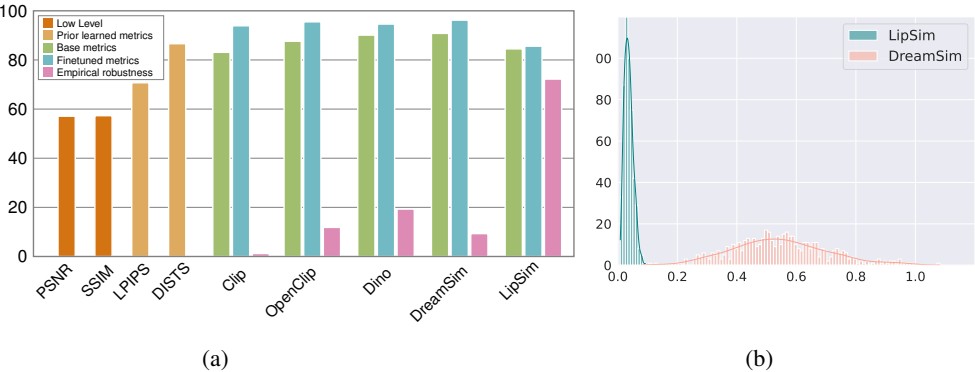

(a)                                                           (b)

Figure 3: Figure 3a compares percentages of alignment of several distance metrics with human vision based on the NIGHT dataset. The ViT-based methods outperform the pixel-wise and CNN-based metrics for the original images. However, LipSim with the 1-Lipschitz backbone composed of CNN and Linear layers has a decent natural score and outperforms the (Base) Clip. Moreover, the figure shows the performance under attack ($\ell_2$-APGD with $\epsilon = 2.0$) for the SOTA metric. While perturbing the reference image, other methods are experiencing a large decay in their performance but LipSim is showing much stronger robustness. Figure 3b shows the distribution of $d(x, x + \delta)$ for LipSim and DreamSim. The $\delta$ perturbation is optimized for each method separately.

## 5.1 VULNERABILITIES OF PERCEPTUAL SIMILARITY METRICS

To investigate the vulnerabilities of DreamSim to adversarial attacks, we aim to answer two questions in this section; Can adversarial attacks against SOTA metrics cause: (1) misalignment with human perception? (2) large changes in distance between perturbed and original images?

**Q1 – Alignment of SOTA Metric with Human Judgments after Attack.** In this part we focus on the binary classification setting and the NIGHT dataset with triplet input. The goal is to generate adversarial attacks and evaluate the resilience of state-of-the-art distance metrics. For this purpose, we use APGD (Croce & Hein, 2020), which is one of the most powerful attack algorithms. During optimization, we maximize the cross-entropy loss, the perturbation $\delta$ is crafted only on the reference image and the two distortions stay untouched:

$$\arg\max_{\delta:\|\delta\|_2 \leq \varepsilon} \mathcal{L}_{ce}(y, \hat{y}) = \mathcal{L}_{ce}([d(x + \delta, x_1), d(x + \delta, x_0)], y) \tag{13}$$

Where $y \in \{0, 1\}$ and $\hat{y} = [d(x + \delta, x_1), d(x + \delta, x_0)]$ which is considered as the logits generated by the model. The natural and adversarial 2AFC scores of DreamSim are reported in Table 1. The natural accuracy drops to half the value for a tiny perturbation of size $\epsilon = 0.5$ and decreases to zero for $\epsilon = 2.0$. In order to visualize the effect of the attack on the astuteness of distances provided by DreamSim, original and adversarial images (that are generated by $\ell_2$-APGD and caused misclassification) are shown in Figure 1. The distances are reported underneath the images as $d(❶, ❷)$. To get a sense of the DreamSim distances between the original and perturbed images, the third row is added so that the original images have (approximately) the same distance to the perturbed images and the perceptually different images in the third row ($d(❶, ❷) = d(❶, ❸)$). The takeaway from this experiment is the fact that tiny perturbations can fool the distance metric to produce large values for perceptually identical images.

**Q2 – Specialized Attack for Semantic Metric.** In this part, we perform a direct attack against the feature extractor model which is the source of the vulnerability for perceptual metrics by employing the $\ell_2$-PGD (Madry et al., 2017) attack ($\epsilon = 1.0$) and the following MSE loss is used during the optimization:

$$\max_{\delta:\|\delta\|_2 \leq \varepsilon} \mathcal{L}_{\text{MSE}}[f(x + \delta), f(x)] \tag{14}$$

The attack is performed on 500 randomly selected samples from the ImageNet-1k test set and against the DreamSim Ensemble feature extractor. After optimizing the $\delta$, the DreamSim distance metric

Table 2: Certified scores of LipSim given different settings. The natural and certified 2AFC scores of all variants of LipSim are shown in this figure. The LipSim - DreamSim version outperforms other variants regarding certified scores. The tradeoff between robustness and accuracy compares the results for different margins in the hinge loss. A higher margin parameter leads to a higher certified score and a lower natural score.

| LipSim with Teacher Model | Margin in Hinge Loss | Natural Score | Certified Score | | |
|---|---|---|---|---|---|
| | | | $\frac{36}{255}$ | $\frac{72}{255}$ | $\frac{108}{255}$ |
| **LipSim – DINO** | 0.2 | 84.76 | 64.14 | 34.76 | 10.53 |
| | 0.4 | 84.65 | 65.19 | 40.51 | 18.04 |
| | 0.5 | 81.96 | 66.28 | 44.63 | 22.49 |
| **LipSim – OpenCLIP** | 0.2 | 83.33 | 61.18 | 34.87 | 13.60 |
| | 0.4 | 80.59 | 63.27 | 42.32 | 21.71 |
| | 0.5 | 81.30 | 64.80 | 45.12 | 25.38 |
| **LipSim – Dreamsim** | 0.2 | **85.58** | 62.88 | 35.36 | 11.18 |
| | 0.4 | 83.33 | 65.40 | 43.69 | 21.11 |
| | 0.5 | 82.89 | **66.39** | **44.90** | **23.46** |

is calculated between the original image and the perturbed image: $d(x, x + \delta)$. The distribution of distances is shown in Figure 3b. We can observe a shift in the mean of the distance from 0 to 0.6 which can be considered as a large value for the DreamSim distance as shown in Figure 1.

## 5.2 LipSim Results

In this section, we aim to leverage the framework introduced in the paper and evaluate the LipSim perceptual metric. In the first step (*i.e.*, right of Figure 2), we train a 1-Lipschitz network for the backbone of the LipSim metric and use the SSL architecture which has 20 Layers of Conv-SSL and 7 layers of Linear-SSL. For training the 1-Lipschitz feature extractor, the ImageNet-1k dataset is used (without labels) and the knowledge distillation approach is applied to utilize the state-of-the-art feature extractors including DINO, OpenCLIP, and DreamSim which is an ensemble of ViT-based models. To enhance the effectiveness of LipSim, we incorporate two parallel augmentation pipelines: standard and jittered. The standard version passes through the feature extractor and the teacher model while the jittered only passes through the feature extractor. Then, the RMSE loss is applied to enforce similarity between the embeddings of the jittered and standard images. This enables LipSim to focus more on the semantics of the image, rather than its colors. After training the 1-Lipschitz backbone of LipSim, we further fine-tune our model on the NIGHT dataset (*i.e.*, step 2 see right of Figure 2). During the fine-tuning process, the embeddings are produced and are projected to the unit $\ell_2$ ball. In order to maintain the margin between logits, the hinge loss is employed similarly to DreamSim. However, while DreamSim has used a margin parameter of 0.05, we used a margin parameter of 0.5 for fine-tuning LipSim in order to boost the robustness of the metric. Remarkably, LipSim achieves strong robustness using a 1-Lipschitz pipeline composed of a 1-Lipschitz feature extractor and a projection to the unit $\ell_2$ ball that guarantees the 1-Lipschitzness of cosine distance. To evaluate the performance of LipSim and compare its performance against other perceptual metrics, we report empirical and certified results of LipSim for different settings.

**Empirical Robustness Evaluation.** We provide the empirical results of LipSim against $\ell_2$-APGD in Table 1. Although the natural score of LipSim is lower than the natural score of DreamSim, there is a large gap between the adversary scores. We can observe that LipSim outperforms all state-of-the-art metrics. The results of a more comprehensive comparison between LipSim, state-of-the-art perceptual metrics, previously proposed perceptual metrics, and pixel-wise metrics are presented in Figure 3a. The pre-trained and fine-tuned natural accuracies are comparable with the state-of-the-art metrics and even higher in comparison to CLIP. In terms of empirical robustness, LipSim demonstrates great resilience. More comparisons

Table 1: Alignment on NIGHT dataset for original and perturbed images using APGD. In this experiment, the perturbation is only applied on the reference images. LipSim demonstrates a higher robustness on all perturbations.

| Metric/ Embedding | Natural Score | $\ell_2$-APGD | | | |
|---|---|---|---|---|---|
| | | 0.5 | 1.0 | 2.0 | 3.0 |
| **CLIP** | 93.91 | 29.93 | 8.44 | 1.20 | 0.27 |
| **OpenCLIP** | 95.45 | 72.31 | 42.32 | 11.84 | 3.28 |
| **DINO** | 94.52 | 81.91 | 59.04 | 19.29 | 6.35 |
| **DreamSim** | **96.16** | 46.27 | 16.66 | 0.93 | 0.93 |
| **LipSim (ours)** | 85.58 | **82.89** | **79.82** | **72.20** | **61.84** |

**LipSim**  **DreamSim**

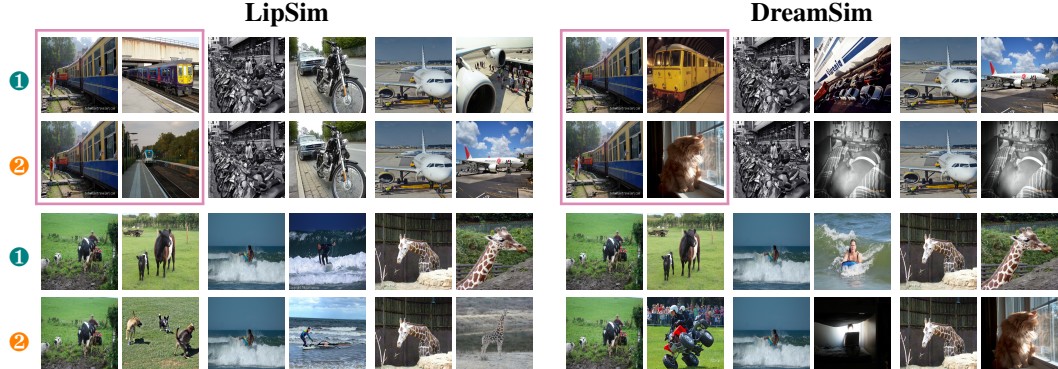

Figure 4: Adversarial attack impact on the performance of DreamSim and LipSim distance metrics on image retrieval application. The ❶ rows show the original images and the top-1 nearest neighbors. Adversarial images generated separately for LipSim and DreamSim metrics along with their top-1 nearest neighbors are depicted in ❷ rows. More precisely, the red block shows a complete sample in the figure, where the upper and lower right images are the original and adversarial queries and the upper and lower left images are the 1-top nearest images to them respectively.

have been performed in this sense, the empirical results over $\ell_\infty$-APGD and $\ell_p$-MIA are also reported in Table 4 and Table 5 in Appendix D which aligns with the $\ell_2$ results and shows strong empirical robustness of LipSim. In order to evaluate the robustness of LipSim outside the classification setting, we have performed $\ell_2$-PGD attack ($\epsilon = 1.0$) using the MSE loss defined in Equation 14 and the distribution of $d(x, x + \delta)$ is shown at Figure 3b. The values of $d(x, x + \delta)$ are pretty much close to zero which illustrates the general robustness of LipSim as discussed in proposition 2. The histogram of the same attack with a bigger perturbation budget ($\epsilon = 3.0$) is shown in Figure 6 of the Appendix.

**Certified Robustness Evaluation.** In order to find the certified radius for data points, the margin between logits is computed and divided by the $\ell_2$ norm distance between embeddings of distorted images ($\|f(x_0) - f(x_1)\|_2$). The results for certified 2AFC scores for different settings of LipSim are reported in Table 2, which demonstrates the robustness of LipSim along with a high natural score. The value of the margin parameter in hinge loss used during fine-tuning is mentioned in the table which clearly shows the trade-off between robustness and accuracy. A larger margin parameter leads to more robustness and therefore higher certified scores but lower natural scores.

## 5.3 Image Retrieval.

After demonstrating the robustness of LipSim in terms of certified and empirical scores, the focus of this section is on one of the real-world applications of a distance metric which is image retrieval. We employed the image retrieval dataset proposed by Fu et al. (2023), which has 500 images randomly selected from the COCO dataset. The top-1 closest neighbor to an image with respect to LipSim and DreamSim distance metrics are shown at the ❶ rows of Figure 4. In order to investigate the impact of adversarial attacks on the performance of LipSim and DreamSim in terms of Image Retrieval application, we have performed $\ell_2$-PGD attack ($\epsilon = 2.0$) with the same MSE loss defined in Equation 14 separately for the two metrics and the results are depicted at the ❷ rows of Figure 4. In adversarial rows, the LipSim sometimes generates a different image as the closest which is semantically similar to the closest image generated for the original image.

## 6 Conclusion

In this paper, we initially showed the vulnerabilities of the SOTA perceptual metrics including DreamSim to adversarial attacks and more importantly presented a framework for training a certifiable robust distance metric called LipSim which leverages the 1-Lipschitz network as its backbone, 1-Lipschitz cosine similarity and demonstrates non-trivial certified and empirical robustness. Moreover, LipSim was employed for an image retrieval task and exhibited good performance in gathering semantically close images with original and adversarial image queries. For future work, It will be interesting to investigate the certified robustness of LipSim for other 2AFC datasets and extend the performance of LipSim for other applications, including copy detection, and feature inversion.

ACKNOWLEDGMENTS

This paper is supported in part by the Army Research Office under grant number W911NF-21-1-0155 and by the New York University Abu Dhabi (NYUAD) Center for Artificial Intelligence and Robotics, funded by Tamkeen under the NYUAD Research Institute Award CG010.

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

## A PROOFS

**Proposition 1.** *Let $f : \mathcal{X} \to \mathbb{R}^k$ be a 1-Lipschitz feature extractor and $\|f(x)\|_2 = 1$, let $d$ be a distance metric defined as in Equation 1 and let $\delta \in \mathcal{X}$ and $\varepsilon \in \mathbb{R}^+$ such that $\|\delta\|_2 \leq \varepsilon$. Then, we have,*

$$|d(x_1, x_2) - d(x_1 + \delta, x_2)| \leq \|\delta\|_2 \tag{3}$$

*Proof of Proposition 1.* We have the following:

$$
\begin{aligned}
|d(x_1, x_2) - d(x_1 + \delta, x_2)| &= \left| \frac{\langle f(x_1 + \delta), f(x_2) \rangle}{\|f(x_1 + \delta)\| \|f(x_2)\|} - \frac{\langle f(x_1), f(x_2) \rangle}{\|f(x_1)\| \|f(x_2)\|} \right| \\
&\overset{(1)}{=} |\langle f(x_1 + \delta), f(x_2) \rangle - \langle f(x_1), f(x_2) \rangle| \\
&= |\langle f(x_1 + \delta) - f(x_1), f(x_2) \rangle| \\
&\leq \|f(x_1 + \delta) - f(x_1)\| \|f(x_2)\| \\
&\overset{(2)}{\leq} \|\delta\|
\end{aligned}
$$

where $(1)$ and $(2)$ are due to $\|f(x)\| = 1$ for all $x$ and the fact that $f$ is 1-Lipschitz, which concludes the proof. $\square$

**Theorem 1** (Certified Accuracy for Perceptual Distance Metric). *Let $H : \mathcal{X} \to \mathbb{R}^2$ be the soft classifier as defined in Equation 4. Let $\delta \in \mathcal{X}$ and $\varepsilon \in \mathbb{R}^+$ such that $\|\delta\|_2 \leq \varepsilon$. Assume that the feature extractor $f : \mathcal{X} \to \mathbb{R}^k$ is 1-Lipschitz and that for all $x$, $\|f(x)\|_2 = 1$, then we have the following result:*

$$M_{H,x} \geq \varepsilon \|f(x_0) - f(x_1)\|_2 \implies M_{H,x+\delta} \geq 0 \tag{7}$$

*Proof of Theorem 1.* First, let us recall the soft classifier $H : \mathcal{X} \to \mathbb{R}^2$ with respect to some feature extractor $f$ as follows:

$$H(x) = [d(x, x_1), d(x, x_0)] \tag{15}$$

where $d : \mathcal{X} \times \mathcal{X} \to \mathbb{R}$ is defined as: $d(x, y) = 1 - \frac{\langle f(x), f(y) \rangle}{\|f(x)\|_2 \|f(y)\|_2}$.

Let us denote $H_0$ and $H_1$ the first and second logits of the soft classifier. For a tuple $(x, x_0, x_1)$ and a target label $y$, we say that $H$ correctly classifies if $\arg \max H(x) = y$. Note that we omit the dependency on $x_0$ and $x_1$ in the notation. Let us assume the target class $y = 1$. The case for $y = 0$ is exactly symmetric. Let us define the margin of the soft classifier $H$ as:

$$M_{H,x} := H_1(x) - H_0(x) \tag{16}$$

We have the following:

$$
\begin{aligned}
M(H, x + \delta) &= H_1(x + \delta) - H_0(x + \delta) \\
&= H_1(x + \delta) - H_1(x) - H_0(x + \delta) + H_0(x) + (H_1(x) - H_0(x)) \\
&= d(x + \delta, x_0) - d(x, x_0) - d(x + \delta, x_1) + d(x, x_1) + (H_1(x) - H_0(x)) \\
&= \left( 1 - \frac{\langle f(x + \delta), f(x_0) \rangle}{\|f(x + \delta)\|_2 \|f(x_0)\|_2} \right) - \left( 1 - \frac{\langle f(x), f(x_0) \rangle}{\|f(x)\|_2 \|f(x_0)\|_2} \right) \\
&\quad - \left( 1 - \frac{\langle f(x + \delta), f(x_1) \rangle}{\|f(x + \delta)\|_2 \|f(x_1)\|_2} \right) + \left( 1 - \frac{\langle f(x), f(x_1) \rangle}{\|f(x)\|_2 \|f(x_1)\|_2} \right) + M_{H,x} \\
&\overset{(1)}{=} -\langle f(x + \delta), f(x_0) \rangle + \langle f(x), f(x_0) \rangle + \langle f(x + \delta), f(x_1) \rangle - \langle f(x), f(x_1) \rangle + M_{H,x} \\
&= \langle f(x + \delta), f(x_1) - f(x_0) \rangle + \langle f(x), f(x_0) - f(x_1) \rangle + M_{H,x} \\
&= \langle f(x + \delta) - f(x), f(x_1) - f(x_0) \rangle + M_{H,x} \\
&\geq -\|f(x) - f(x + \delta)\| \|f(x_0) - f(x_1)\| + M_{H,x} \\
&\geq -\varepsilon \|f(x_0) - f(x_1)\| + M_{H,x}
\end{aligned}
$$

where $(1)$ is due to the fact that for all $x$, $\|f(x)\| = 1$. Therefore, $M_{H,x+\delta} \geq 0$ only if $M_{H,x} \geq \varepsilon \|f(x_0) + f(x_1)\|$ which conclude the proof. $\square$

## B    DATASET DETAILS & ADDITIONAL FIGURES

In the section, we initially represent some details of 2AFC datasets and more specifically NIGHT dataset and show some examples of this dataset.

### B.1    DATASET DETAILS

In order to train a perceptual distance metric, datasets with perceptual judgments are used. The perceptual judgments are of two types: two alternative forced choice (2AFC) test, that asks which of two variations of a reference image is more similar to it. To validate the 2AFC test results, a second test, just a noticeable difference (JND) test is performed. In the JND test, the reference images and one of the variations are asked to be the same or different. BAPPS (Zhang et al., 2018) and NIGHT (Fu et al., 2023) are two datasets organized with the 2AFC and JND judgments. The JND section of the NIGHT dataset has not been released yet, therefore we did our evaluations only based on the 2AFC score. In Figure 5 we show some instances from the NIGHT dataset, the reference is located in the middle and the two variations are left and right. The reference images are sampled from well-known datasets including ImageNet-1k.

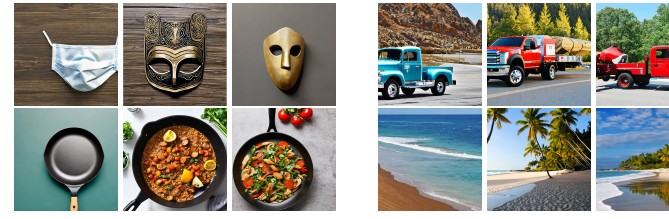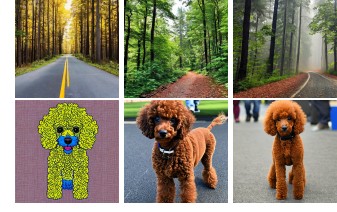

Figure 5: Some instances of NIGHT dataset is shown. The middle image is the reference image and right and left images are $x_0$ and $x_1$ distortions respectively.

## C    CERTIFIED ROBUSTNESS ON BAPPS DATASET

In this section, we aim to show the generalization of LipSim for other 2AFC dataset available called BAPPS (Zhang et al., 2018). The BAPPS has the same structure with NIGHT dataset as explained at Appendix B. However, the left and right images have been generated by adding distortions to the reference image and the label determines which of the distortions is more similar to the reference. Therefore the distribution of data for NIGHT and BAPPS datasets are totally different. To provide certification for the BAPPS dataset, the labels which are are between 0 and 1 are rounded to have binary labels for all samples. Later, we leverage the finetuned verions of LipSim to provide certificates on the BAPPS dataset. Although LipSim is fine-tuned on the NIGHT dataset and the distribution of images differ for these two datasets, it has good certified scores on BAPPS dataset as reported at Table 3.

Table 3: Certified scores of LipSim on BAPPS dataset.

| Metric | Margin in Hinge Loss | Natural Score | Certified Score | | |
|---|---|---|---|---|---|
| | | | $\frac{36}{255}$ | $\frac{72}{255}$ | $\frac{108}{255}$ |
| **DreamSim** | - | 78.47 | 0.0 | 0.0 | 0.0 |
| **LipSim** | 0.2 | 73.47 | 30.0 | 12.96 | 5.33 |
| | 0.4 | 74.31 | 31.74 | 15.19 | 7.0 |
| | 0.5 | 74.29 | 31.20 | 15.07 | 6.77 |

# D    ADDITIONAL EMPIRICAL RESULTS FOR LIPSIM

In this section we aim to perform a diverse set of empirical analysis on LipSim. At first we revisit the empirical robustness of LipSim using different adversarial attacks. Second we compare LipSim with a recent empirical defense called R-LPIPS Ghazanfari et al. (2023) on both NIGHT and BAPPS dataset. Later we look into the general robustness property and perform stronger attack directly to the metric. Afterwards, we present another application for LipSim as a perceptual metric which is the KNN Task and Finally we have the comparison between LipSim and other metrics in terms of natural score.

## D.1    EMPIRICAL ROBUSTNESS RESULTS

In this part, the empirical robustness of LipSim is evaluated using $\ell_\infty$-APGD, $\ell_\infty$-PGD attack and Momentum Iterative Attack Dong et al. (2018). The cross entropy loss as defined in Equation 13 is used for the optimization. In the case of $\ell_\infty$-APGD, LipSim outperforms all metrics for the entire set of perturbation budgets. For $\ell_\infty$-PGD, the performance of DreamSim is better for $\epsilon = 0.01$, however, LipSim has kept a high stable score encountering PGD attacks with different perturbation sizes, which demonstrates the stability of the LipSim. The results for Momentum Iterative Attack are reported at Table 5, which is in line with the other results presented in the paper in terms of empirical robustness of LipSim towards adversarial attacks.

Table 4: Alignment on NIGHT dataset for original and perturbed images using APGD. In this experiment, different feature extractors are employed in combination with cosine distance to calculate the distance between reference images and distorted images, and the perturbation is only applied to the reference images. DINO, CLIP, and OpenCLIP are ViT-based feature extractors, DreamSim Ensemble is a concatenation of all these three feature extractors and LipSim Backbone is a 1-Lipschitz network that is trained from scratch using the knowledge distillation approach.

| Metric/ Embedding | Natural Score | $\ell_\infty$-APGD | | |
|---|---|---|---|---|
| | | 0.01 | 0.02 | 0.03 |
| **DINO** | 94.52 | 14.91 | 1.42 | 0.21 |
| **CLIP** | 93.91 | 0.93 | 0.05 | 0.00 |
| **OpenCLIP** | 95.45 | 7.89 | 0.76 | 0.05 |
| **DreamSim** | **96.16** | 2.24 | 0.10 | 0.05 |
| **LipSim** | 85.58 | **62.28** | **33.66** | **15.19** |

Table 5: Alignment on NIGHT dataset for original and perturbed images using Momentum Iterative Attack.

| Metric | Natural Score | $\ell_2$-MIA | | | $\ell_\infty$-MIA | | |
|---|---|---|---|---|---|---|---|
| | | 0.5 | 1 | 2 | 0.01 | 0.02 | 0.03 |
| **DreamSim** | **96.16** | 61.79 | 52.85 | 52.69 | 2.08 | 0.05 | 0.0 |
| **LipSim** | 85.58 | **82.79** | **79.99** | **80.10** | **62.45** | **34.38** | **15.84** |

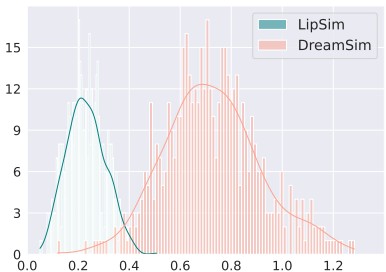

Figure 6: Distribution of $d(x, x + \delta)$ where $\delta$ is generated using $\ell_2$-PGD attack with $\epsilon$=3.0

## D.2 GENERAL ROBUSTNESS OF LIPSIM

In order to evaluate the general robustness of LipSim in comparison with DreamSim, we optimize the MSE loss defined in Equation 14 employing $\ell_2$-PGD attack for LipSim and DreamSim separately. The distribution of $d(x, x + \delta)$ is shown in Figure 6. The difference between this figure and the histogram in Figure 3b is that we have chosen a larger perturbation budget, $\epsilon = 3$, to demonstrate the fact that even for larger perturbations, LipSim is showing general robustness.

## D.3 COMPARISON WITH AN EMPIRICAL DEFENSE

As discussed in the Related Work, there exist a few papers about providing defense for perceptual metrics. R-LPIPS (Ghazanfari et al., 2023) is a recent paper to propose an empirical defense for the LPIPS (Zhang et al., 2018) perpetual similarity metric by leveraging Adversarial training and $\ell_\infty$-PGD attacks. For our evaluation we perform $\ell_\infty$-PGD and $\ell_2$-MIA (Momentum Iterative attack) and provide an empirical robustness score on the BAPPS dataset as well as the NIGHT dataset in Table 6. Finally we need to emphasize that R-LPIPS is trained on the BAPPS dataset while Lipsim is finetuned on the NIGHT dataset and Lipsim provides certified scores while R-LPIPS does not come with any provable guarantees.

Table 6: Alignment on NIGHT and BAPPS dataset for original and perturbed images with LipSim and R-LPIPS metrics.

| Metric | Dataset | **Natural Score** | $\ell_2$-**MIA** $\epsilon = 1.0$ | $\ell_\infty$-**PGD** $\epsilon = 0.03$ |
|--------|---------|-------------------|-----------------------------------|------------------------------------------|
| **R-LPIPS** | NIGHT | 70.56 | 58.50 | 32.46 |
|             | BAPPS | 80.25 | 72.38 | 70.94 |
| **LipSim** | NIGHT | 85.58 | 79.99 | 75.27 |
|            | BAPPS | 73.47 | 60.09 | 42.77 |

## D.4   MORE APPLICATIONS: KNN TASK

For the applications of LipSim, we have presented image retrieval in Section 5 and for a second application, we add the KNN task to this part. KNN (k-nearest neighbors algorithm) is a zero-shot classification task that classifies test images based on the proximity of their feature vectors to the training images' feature vectors. We performed our experiment on Imagenette[2] (ImageNet dataset with 10 classes) and for $k = \{10, 20\}$. The accuracy of LipSim and DreamSim for the KNN task are reported in Table 7. LipSim is providing more than $85\%$ accuracy on classification task by leveraging its robust embeddings and its accuracy is also very close to DreamSim in terms of Top 5.

Table 7: LipSim and DreamSim comparison for the KNN task on Imagenette dataset.

| Metric | 10-NN | | 20-NN | |
| --- | --- | --- | --- | --- |
| | Top 1 | Top 5 | Top 1 | Top 5 |
| **DreamSim** | 99.03 | 99.82 | 98.82 | 99.89 |
| **LipSim** | 85.32 | 97.20 | 85.35 | 98.09 |

---

[2]https://github.com/fastai/imagenette

