# OpenReview forum: "LipSim: A Provably Robust Perceptual Similarity Metric"
_ICLR.cc/2024/Conference — ICLR 2024 poster_

### Official Review · Reviewer_sTNV · 2023-11-01

**Soundness:** 3 good
**Presentation:** 3 good
**Contribution:** 3 good
**Rating:** 6
**Confidence:** 3

**Summary:**

Authors argue that perceptual metrics inherit both the strengths and shortcomings of neural networks. One of the important shortcomings of neural Networks is their vulnerability to adversarial examples.
Thus it makes sense to design adversarially robust nn-based perceptual metrics. They show that the DreamSim metric is not robust to adversarial examples. They then propose a robust perceptual
similarity metric called LipSim (Lipschitz Similarity Metric) with provable guarantees.

**Strengths:**

The presented method is first in its kind to provide a provably robust perceptual metric. Authors have shown that this method is more robust than existing perceptual metrics (DreamSim). Several experiments are conducted to show the efficacy of the method.

The paper in general is well organized and well-written. There are also several experiments that show effectiveness of the method.

**Weaknesses:**

Although there has been some previous works on robust perceptual metrics, authors claim theirs is the first one with provable guarantees. I still think discussing why this matters in practice is important. I see the application for image retrieval, but how can someone have access to the model (white box) to actually attack it. Please elaborate.

Consider changing the colors in bar plot of Fig. 3.a. They are hard to distinguish.

**Questions:**

Q: Please mention why we need robust perceptual metrics??!!! In real world. In other words, how much impact such work has in real world and why it actually matters? An example may help here.

Q: How about some non NN-based similarity metrics? I am guessing maybe there are not as good as NN-based ones but at least they might be more robust! Can you compare your method agains some of those m
Metrics? Is DISTS in Fig. 3.a NN-based?

Q: Other datasets than Nights dataset? What is the guarantee that these results will also generalize to other datasets?


Q: In 5.1, you are stating “Can adversarial attacks against SOTA metrics cause: (1) misalignment with human perception?”.
In section 4.1, you mention “Q: Please mention why we need robust perceptual metrics??!!! In real world. In other words, how much impact such work has in real world and why it actually matters? An example may help here. Alternatively, shouldn’t two image differ significantly when perturbation is high? Have you considered what happens at high perturbations?
Shouldn’t a good perceptual metric match human judgements regardless of perturbation magnitude?


Q: Table 2 shows the certified scores. However, the scores for perturbation 72/255 and beyond are still pretty low. How do humans behave in those perturbations?

---

> ### Author Response · Authors · 2023-11-17
> **Response to Reviewer sTNV**
>
> We thank the reviewer for their valuable feedback and comments.
>
> **Q: Although there has been some previous works on robust perceptual metrics, authors claim theirs is the first one with provable guarantees. I still think discussing why this matters in practice is important.**
>
> We provide the first perceptual similarity metric with non-trivial provable guarantees. We do agree that comparing to existing work is important and we have added the comparison with other metrics to the appendix of the revision of our paper and provided the reference to it in the Related Work.
>
> There exist two methods that apply randomized smoothing to perceptual similarity metrics [1, 2]. We have mentioned them in the related work section of our paper. These methods have important limitations:
> - They are computationally _very expensive_ due to the Monte Carlo sampling for each data point (i.e., [1] mentions 1 to 3 minutes for each image with a backbone smaller than the Dreamsim model) which makes this approach impractical for real use cases with a large model like Dreamsim
> - The approach is probabilistic due to the estimation of the probability with the Monte Carlo sampling
> - The proposed certified bounds are loose, for example, for LPIPS [6] the bound is as follows: $d(x, x+\delta) \leq 10 ||\delta||_2$. Therefore the results are reported for a perturbation budget of 0.01 to 0.05, whereas for the LipSim the certificates are provided for a perturbation budget of $\frac{36}{255}$ to $\frac{108}{255}$.
>
> **Q: I see the application for image retrieval, but how can someone have access to the model (white box) to actually attack it. Please elaborate.**
>
> In security research, it is common practice to study the worst-case setting, i.e., white-box attacks in the context of adversarial robustness. This is for multiple reasons:
> - (1) in practice, many neural networks are released in white-box open form, for example, the most recent models from Meta. These networks do in fact need to be robust to adversarial attack.
> - (2) Even black-box models might need to consider vulnerability to white-box attacks in case they are inadvertently leaked (data breaches are a common occurrence). Relying on a lack of access to the details of an implementation, i.e., security via obscurity, is considered as a bad practice in the literature.
> - (3) Finally, several results show that black-box models can be attacked using _transferability_ [3]  i.e., attacks obtained on proxy models often work just as well on other models.
>
> In this paper, we provide robustness in a white-box setting, specifically with certified robustness, and show that we can provide security and reliability within the certified bounds and we also evaluate our model with empirical approaches.
>
> **Q: Consider changing the colors in bar plot of Fig. 3.a. They are hard to distinguish.**
>
> Sure, based on your suggestion, we changed the colors of Fig. 3.a in the revision of our paper.

---

> ### Author Response · Authors · 2023-11-17
>
> **Q: Please mention why we need robust perceptual metrics??!!! In real world. In other words, how much impact such work has in real world and why it actually matters? An example may help here.**
>
> Perceptual metrics have many different applications in the real world, namely, the comparison of images using a human semantic measure for image and video processing, image retrieval based on semantic information, and filter detection in social media websites (e.g., CSAM$^1$ detection see below). These contexts can often be in an adversarial setting $^2$ and having a perceptual metric robust to small perturbation, i.e., perturbations that do not change the semantic, is a property that makes sense to have in this type of metric.
> We should note that perceptual metrics can also be used to train networks which requires a semantic loss, having a robust perceptual metric could lead to better and more robust models.
>
> In this work, we leveraged the DreamSim metric [4] which is the state-of-the-art for perceptual similarity metric $^3$. While this metric provides a great similarity measure, we have demonstrated that it is not robust to small perturbations *that do not change the semantic* (As shown in Figure 1 of the paper, a small perturbation can change the perception of the Dreamsim metric, and two images semantically similar are then considered very different.). If this metric were to be deployed into real-world applications, this might lead to important security issues.
>
> As an example of a real-world application, we could mention Apple’s NeuralHash [5] used for CSAM detection, which has two steps: (1) it uses a Convolutional Neural Network to map semantically similar images to close feature vectors (in terms of cosine similarity) and (2) it leverages those vectors to find the duplicates using hash functions. Based on the pipeline, the vulnerability of the NN generating the feature vectors is a serious security concern. We refer to [6] as one of the papers that worked on the adversarial vulnerabilities of the Neural Network generating the features for the NeuralHash and showed that using the proposed adversarial attack, the features can change and lead to a different output for the hash function. It would be a very interesting future work to use the robust embeddings we've trained as the backbone of NeuralHash to increase the robustness of the feature vectors.
>
> $^1$ CSAM: Child Sexual Abuse Material
> $^2$ CSAM detection is inherently an adversarial setting where malicious users would want to bypass the detection.
> $^3$ The Dreamsim paper has been accepted at NeurIPS 2023 as a spotlight.
>
>
> **Q: How about some non NN-based similarity metrics? I am guessing maybe there are not as good as NN-based ones but at least they might be more robust! Can you compare your method against some of those Metrics? Is DISTS in Fig. 3.a NN-based?**
>
> We have reported the results for the non-NN-based metrics with the "Low level" title in Figure 3.A. The natural 2AFC scores are low and are not comparable with Dreamsim. This result is expected because the 2AFC score shows the alignment of the distance metric with human perception and how well the metric compares the images at the semantic level. Pixel-level metrics are more focused on the pixel level rather than the semantic level and therefore less aligned with human perception in comparison with the perceptual metrics which are more focused on the semantics. On the other hand, as the natural scores of non-NN-based metrics are already lower than LipSim's robust scores (under $\ell_2$-AutoAttack) (and their robust scores won't be better than their natural scores,) we didn't need to do the experiments under the attack for the "Low level" methods.
>
> DISTS indeed offers competitive performance on the NIGHT dataset as shown in Figure 3.a. The DISTS[7] metric is based on the VGG architecture (like the LPIPS is based in the ALexNet architecture) and due to being based on a neural network is not robust to small perturbations crafted using a strong attack. However, the certified robustness of LipSim guarantees that independent from the strength of the attack, the certified scores cannot be broken.
>
> | Metric | Accuracy | $\frac{36}{255}$ | $\frac{72}{255}$ | $\frac{108}{255}$ |
> | ---- | --- |---- | ------ | ------ |
> | DISTS | 86.0 | 0.0 | 0.0 | 0.0 |
> | LipSim | 85.09 | 67.32 | 43.26 | 19.02 |

---

> ### Author Response · Authors · 2023-11-17
>
> **Q: Other datasets than NIGHTs dataset? What is the guarantee that these results will also generalize to other datasets?**
>
> Currently, two 2AFC datasets are available: NIGHT and BAPPS [8]. The result of LipSim on the NIGHT dataset is reported in the paper. Here we report the result of LipSim on the BAPPS dataset. The LipSim model is not finetuned on the BAPPS dataset and the same model which is finetuned on the NIGHT dataset is used.
>
> | Metric | Accuracy | $\frac{36}{255}$ | $\frac{72}{255}$ | $\frac{108}{255}$ |
> | ---- | --- |---- | ------ | ------ |
> | DreamSim | 78.47 | 0.0 | 0.0 | 0.0 |
> | LipSim | 74.29 | 31.20 | 15.07 | 6.77 |
>
> **Q: (1) Alternatively, shouldn’t two images differ significantly when perturbation is high? (2) Have you considered what happens at high perturbations? (3) Shouldn’t a good perceptual metric match human judgements regardless of perturbation magnitude? (4) Table 2 shows the certified scores. However, the scores for perturbation 72/255 and beyond are still pretty low. How do humans behave in those perturbations?**
>
> 1 - Yes, two images should differ significantly when the perturbation is high.
> 2 - We have experimented with increasing perturbation, starting with a small perturbation $\epsilon = \frac{36}{255}$ and increasing to $\epsilon = \frac{108}{255}$. As shown in the paper, perceptual similarity metrics (as with neural networks) are not robust to small adversarial perturbations. It is therefore safe to say that won't be robust to large adversarial perturbations.
> 3 - Yes, a good perceptual metric should match human judgments regardless of perturbation magnitude. This highlights the fact that building a good perceptual metric is still an open problem and our contribution is a first step in this direction. Our Lipsim perceptual metric achieves competitive results on natural performance and obtains a good level of robustness that Dreamsim doesn't have.
> 4 - The perturbation level of $\epsilon = \frac{72}{255}$ is classic in the adversarial robustness literature, for example, the state-of-the-art certified robust classifier with Lipschitz continuity [9] reports a certified accuracy with $\epsilon = \frac{36}{255}$.
>
> [1] Aounon Kumar and Tom Goldstein. Center smoothing: Certified robustness for networks with structured outputs. NeurIPS 2021
> [2] Shao et al. Robustness certification for structured prediction with general inputs via safe region modeling in the semimetric output space. SIGKDD 2023
> [3] Madry, Aleksander, et al. "Towards deep learning models resistant to adversarial attacks." arXiv preprint arXiv:1706.06083 (2017).
> [4] Fu et al. DreamSim: Learning New Dimensions of Human Visual Similarity using Synthetic Data, NeurIPS 2023 Spotlight.
> [5] Apple. CSAM Detection - Technical Summary. https://www.apple.com/childsafety/pdf/CSAM_Detection_Technical_Summary.pdf 2021.
> [6] Struppek et al. "Learning to break deep perceptual hashing: The use case neuralhash." ACM Conference on Fairness, Accountability, and Transparency. 2022.
> [7] Ding et al. Image quality assessment: Unifying structure and texture similarity. IEEE transactions on pattern analysis and machine intelligence 2020.
> [8] Richard Zhang, Phillip Isola, Alexei A Efros, Eli Shechtman, and Oliver Wang. The unreasonable effectiveness of deep features as a perceptual metric. In Proceedings of the IEEE conference on computer vision and pattern recognition, 2018.
> [9] Hu et al. Unlocking Deterministic Robustness Certification on ImageNet. NeuIPS 2023.

---

### Official Review · Reviewer_YdcH · 2023-11-01

**Soundness:** 3 good
**Presentation:** 3 good
**Contribution:** 3 good
**Rating:** 5
**Confidence:** 2

**Summary:**

This study investigates Lipschitz Networks' robustness against small adversarial perturbations, capitalizing on Lipschitz continuity. A 1-Lipschitz function can bound a network's output to prevent it from changing excessively in response to slight input changes. Another key focus is the DreamSim perceptual distance metric, which utilizes cosine distance on combined feature vectors derived from various representation learning techniques. Furthermore, a new metric, LipSim, is introduced, aiming to be robust against adversarial disturbances. LipSim's design employs a two-step training process that first uses DreamSim as a guiding model on the ImageNet dataset, followed by fine-tuning on the NIGHT dataset to enhance its robustness.

**Strengths:**

- LipSim employs a 1-Lipschitz network backbone, which when combined with certain design choices, greatly enhances its resistance to adversarial perturbations.
- When applied to the real-world task of image retrieval, LipSim effectively identified semantically close images, even when faced with adversarial image queries.
-  LipSim excels both in empirical and certified robustness tests. This dual proficiency ensures that the metric's performance is not only observed in experimental conditions but also certified under specific robustness criteria.

**Weaknesses:**

- The natural score of LipSim was observed to be lower than that of some competitors like DreamSim. This might raise concerns about its general performance when not under adversarial conditions.
- The real-world application testing of LipSim was primarily on image retrieval. It would be beneficial to see its performance on a wider variety of tasks.

**Questions:**

n/a

**Details Of Ethics Concerns:**

No concern

---

> ### Author Response · Authors · 2023-11-17
> **Response to Reviewer YdcH**
>
> We thank the reviewer for their valuable feedback and comments.
>
> **Q: The natural score of LipSim was observed to be lower than that of some competitors like DreamSim. This might raise concerns about its general performance when not under adversarial conditions.**
>
> This paper is set in the context of adversarial robustness. In this context, there is an _inherent trade-off between natural performance and robustness_ $^1$. State-of-the-art methods, like DreamSim, can achieve high natural performance at the expense of losing robustness, as demonstrated in Figure 1 of our paper. LipSim does not achieve state-of-the-art natural performance but achieves competitive performance compared to Dreamsim _and_ provides state-of-the-art empirical and certified robustness. We believe Lipsim is a major first step towards robust perceptual similarity metrics.
>
> $^1$ As an example, the state-of-the-art robust neural network (accepted at NeurIPS 2023) [1] with Lipschitz continuity achieves 45.6% natural accuracy on ImageNet with 35.0% certified accuracy while the non-robust state-of-the-art results on ImageNet is 91.1% [2] and the community has been trying to boost the natural accuracy for robust models, despite the progressions this is still an ongoing topic of research.
>
>
> **Q: The real-world application testing of LipSim was primarily on image retrieval. It would be beneficial to see its performance on a wider variety of tasks.**
>
> Based on your suggestion, we are also adding KNN (k-nearest neighbors algorithm), which is a zero-shot classification task. It classifies test images based on the proximity of their feature vectors to the training images' feature vectors. We performed our experiment on Imagenette (ImageNet dataset with 10 classes) [3] and for $k=10,20$. The results of LipSim and DreamSim are reported in the following table, which are very close in terms of the Top 5 and LipSim is showing a decent performance in terms of the Top 1 as a zero-shot classifier.
>
> | Metric | 10-NN Top 1 | 10-NN Top 5 | 20-NN  Top 1 | 20-NN Top 5 |
> | ---- | --- |---- | ------ | ------ |
> | DreamSim | 99.03 | 99.82 | 98.82 | 99.89 |
> | LipSim | 85.32 | 97.20 | 85.35 | 98.09 |
>
> [1] Hu et al. Unlocking Deterministic Robustness Certification on ImageNet. NeuIPS 2023
> [2] Chen et al. Symbolic Discovery of Optimization Algorithms. NeuIPS 2023
> [3] https://github.com/fastai/imagenette

---

### Official Review · Reviewer_KFDY · 2023-11-02

**Soundness:** 2 fair
**Presentation:** 2 fair
**Contribution:** 2 fair
**Rating:** 5
**Confidence:** 2

**Summary:**

This paper studies how to develop a robust perceptual similarity metric. Specifically, this paper propose to leverage 1-Lipschitz neural network to provide guarded areas around each data point. The experimental results illustrate the effectiveness of the proposed method.

**Strengths:**

The experimental results looks very good when against auto attacks.

The proof seems to make sense but I am not an export with this.

**Weaknesses:**

The presentation could be better. For example, the explanation of 2AFC is little which making it difficult to get the messages from this paper.

The experiments only conducted with Auto Attack. However, there are different kinds of attacks, and it would be good to experiment with other attack methods as well.

It would also be good to compare with other certified or non-certified defense methods.

**Questions:**

See weakness.

---

> ### Author Response · Authors · 2023-11-17
> **Response to Reviewer KFDY**
>
> We thank the reviewer for their valuable feedback and comments.
>
> **Q: The presentation could be better. For example, the explanation of 2AFC is little which making it difficult to get the messages from this paper.**
>
> We had provided a detailed explanation of 2AFC datasets in Appendix B.1 but did not reference it in the introduction. We have corrected this in the revised version of the paper.
>
> Conceptually, 2AFC datasets are datasets labeled by humans that are used to align perceptual metrics to human judgment. The dataset is labeled via a two-alternative forced choice (2AFC) test that asks which of two variations of a reference image is more similar to it. We provided examples of the night dataset in Appendix B.1 and gave more detail on how the dataset is constructed. For a full description of this dataset, please see the DreamSim paper [1].
>
>
> **Q: The experiments only conducted with Auto Attack. However, there are different kinds of attacks, and it would be good to experiment with other attack methods as well.**
>
> We agree that it is important to evaluate with a diversity of different attacks. Indeed, AutoAttack [2] is itself an ensemble of four different white box and black box attacks (i.e. APGD-CE, APGD-DLR, with two existing complementary attacks, FAB [3] and Square Attack [4]). We have further clarified and emphasized this in the revised paper. By choosing AutoAttack, we have both the state-of-the-art attack and the diversity of white box and black box attacks as you mentioned. Based on your input, we have further increased attack diversity by adding results from two additional attacks as explained below.
>
> For our experiment, we evaluate the LipSim in two different settings:
>
> **1 - NIGHT dataset (2AFC dataset)**:  the night dataset is a binary classification setting and the attacker's goal is to make the model misclassify. The following attack methods are used to evaluate the robustness of LipSim in the 2AFC setting:
>
> - We provided $\ell_2$-AutoAttack results in Table 1 of the paper. AutoAttack [2] is known as a state-of-the-art attack in the adversarial robustness community.
> - We also have provided the results for the $\ell_{\infty}$-AutoAttack and $\ell_{\infty}$-PGD shown in Table 3 of the Appendix.
> - Based on your suggestion regarding the evaluation of LipSim using different attacks, we added the Momentum Iterative Attack (MIA) [5] for both $\ell_{\infty}$ and $\ell_{2}$. The new results are in line with our previous results and show the superiority of LipSim compared with DreamSim in terms of robustness.
> - We also added the results on another dataset per the suggestion of Reviewer sTNV.
>
> | Metric | Natural Score| $\ell_2$-MIA $\epsilon= 0.5$ | $\ell_2$-MIA $\epsilon= 1.0$ | $\ell_2$-MIA $\epsilon= 2.0$ | $\ell_{\infty}$-MIA $\epsilon=0.01$ | $\ell_{\infty}$-MIA $\epsilon=0.02$ | $\ell_{\infty}$-MIA $\epsilon=0.03$ |
> | ---- | ------ | --- | ---- | ------ | --- | ---- | ------ |
> | DreamSim | 96.16 | 61.79 | 52.85 | 52.69 | 2.08 | 0.05 | 0.0 |
> | LipSim | 85.09 | 82.79 | 79.99 | 80.10 | 62.45 | 34.38 | 15.84|
>
> **2 - Perceptual metric**: In this setting, we evaluate the robustness of Dream and LipSim (denoted $d$) using a direct attack to the metric by optimizing the perturbation within a budget in the loss defined in Equation 14 $\underset{||\delta||_2 \leq \epsilon}{argmax}\, d(x, x+\delta)$, we demonstrated that we can find small perturbations that fool the metric to predict larger values for $d(x, x+\delta)$.
> The distribution $d(x, x+\delta)$ is shown in Figure 3b, for both LipSim and DreamSim, which highlights the robustness gaps between LipSim and DreamSim.

---

> ### Author Response · Authors · 2023-11-17
>
> **Q: It would also be good to compare with other certified or non-certified defense methods.**
>
> **Certified Defenses**: There exist two methods that apply randomized smoothing to perceptual similarity metrics [6, 7]. We mentioned them in the related work section of our paper. These methods have important limitations:
> - They are computationally *very expensive* due to the Monte Carlo sampling for each data point (i.e., [6] mentions 1 to 3 minutes for each image with a backbone smaller than the Dreamsim model) which makes this approach impractical for real use cases with a large model like Dreamsim
> - The approach is probabilistic due to the estimation of the probability with the Monte Carlo sampling
> - The proposed certified bounds are loose, for example, for LPIPS [6] the bound is as follows: $d(x, x+\delta) \leq 10 ||\delta||_2$. Therefore the results are reported for a perturbation budget of 0.01 to 0.05, whereas for the LipSim the certificates are provided for a perturbation budget of $\frac{36}{255}$ to $\frac{108}{255}$.
> We have highlighted the difference between Lipsim and previous approaches in the revision of the paper.
>
> **Non-certified Defenses (Empirical robustness)**: R-LPIPS [8] leverages the LPIPS perpetual similarity metric and Adversarial training to provide empirical robustness. We provided an empirical robustness score on the BAPPS dataset [9] as well as the NIGHT dataset [1] below. We would like to emphasize that R-LPIPS is trained on the BAPPS dataset while Lipsim is finetuned on the NIGHT dataset and Lipsim provides certified accuracy while R-LPIPS does not come with any provable guarantees.
>
> - Results on the BAPPS dataset
>
> | Metric | Natural Accuracy | $\ell_{\infty}$-PGD  $\epsilon=0.03$ | $\ell_{2}$-MIA $\epsilon=1.0$  |
> | ---- | --- | --- | --- |
> | R-LPIPS | 80.25 | 70.94 | 72.38 |
> | LipSim | 73.47 | 42.77 | 60.09 |
>
> - Results on the NIGHT dataset
>
> | Metric | Natural Accuracy | $\ell_{\infty}$-PGD  $\epsilon=0.03$ | $\ell_{2}$-MIA $\epsilon=1.0$ |
> | ---- | --- | --- | --- |
> | R-LPIPS | 70.56 | 32.46 | 58.50 |
> | LipSim | 85.09 | 75.27 | 79.99 |
>
>
> [1] Fu et al. DreamSim: Learning New Dimensions of Human Visual Similarity using Synthetic Data, NeurIPS 2023 Spotlight.
> [2] Croce et al. Reliable Evaluation of Adversarial Robustness with an Ensemble of Diverse. ICML 2020.
> [3] Croce, F. and Hein, M. Minimally distorted adversarial examples with a fast adaptive boundary attack. In ICML, 2020.
> [4] Andriushchenko, M., Croce, F., Flammarion, N., and Hein,
> M. Square attack: a query-efficient black-box adversarial
> attack via random search. In ECCV, 2020.
> [5] Dong et al. Boosting Adversarial Attacks with Momentum. CVPR 2018
> [6] Kumar et al. Center smoothing: Certified robustness for networks with structured outputs. NeurIPS 2021
> [7] Shao et al. Robustness certification for structured prediction with general inputs via safe region modeling in the semimetric output space. SIGKDD 2023
> [8] Ghazanfari et al. R-lpips: An adversarially robust perceptual similarity metric. ICML Workshop 2023.
> [9] Richard Zhang, Phillip Isola, Alexei A Efros, Eli Shechtman, and Oliver Wang. The unreasonable
> effectiveness of deep features as a perceptual metric. In Proceedings of the IEEE conference on computer vision and pattern recognition, 2018.

---

### Author Response · Authors · 2023-11-17
**General Response**

We would like to thank the reviewers for their interest, positive remarks, and comments. Below we emphasize two important comments on the contributions of our work and provide a summary of the changes in the revision of our paper. Please feel free to reach out if there are still questions or comments you would like us to address.

### **1 - Motivation -- On the importance of having a _robust_ perceptual metric**

Perceptual metrics have many different applications in the real world, namely, the comparison of images using a human semantic measure for image and video processing, retrieving images based on semantic information, and identifying or detecting inappropriate/adult content in social media websites (e.g., CSAM$^1$ detection, as described below). These contexts can often be in an adversarial setting $^2$ and having a perceptual metric robust to small perturbation, i.e., perturbations that do not change the semantics, is an essential property to have in this type of metric.
We should note that perceptual metrics can also be used to train networks that require a semantic loss; therefore, a robust perceptual metric could lead to better and more robust models.

In this work, we leveraged the DreamSim metric [1] which is the state-of-the-art for perceptual similarity $^3$. While this metric provides a strong similarity measure, we have demonstrated that it is not robust to small perturbations *that do not change the image semantics* (As shown in Figure 1 of the paper, a small perturbation can change the perception of the Dreamsim metric, *i.e.*, two images semantically similar are then considered very different.). If this metric were to be deployed in real-world applications, it may lead to important security issues.

As an example of a real-world application, we highlight Apple’s NeuralHash [4] used for CSAM detection, which has two steps: (1) it uses a Convolutional Neural Network (CNN) to map semantically similar images to close feature vectors (in terms of cosine similarity) and (2) it leverages those vectors to find duplicates using hash functions. Hence, the vulnerability of the CNN generating the feature vectors is a serious security concern. For example, [5] has proposed an adversarial attack that changes CNN features such that their hash function outputs are very different, thus evading detection. It would be interesting for future work to use the robust embeddings we have trained as the backbone of NeuralHash to increase the robustness of the feature vectors.

$^1$ CSAM: Child Sexual Abuse Material
$^2$ CSAM detection is inherently an adversarial setting where malicious users would want to bypass the detection.
$^3$ The Dreamsim paper has been accepted at NeurIPS 2023 as a spotlight.

### **2 - On the significance of our experiments**

This paper is set in the context of adversarial robustness. In this context, there is an _inherent trade-off between natural performance and robustness_ $^1$. State-of-the-art methods, like DreamSim, can achieve good natural performance at the expense of losing robustness, as demonstrated in Figure 1 of our paper. LipSim does not achieve state-of-the-art natural performance but achieves competitive performance compared to Dreamsim _and_ provides state-of-the-art empirical and certified robustness. We believe Lipsim is a major first step towards robust perceptual similarity metrics.

$^1$ As an example, the state-of-the-art robust neural network (NeurIPS 2023) [2] with Lipschitz continuity achieves 45.6% natural accuracy on ImageNet with 35.0% certified accuracy while the non-robust state-of-the-art results on ImageNet is 91.1% [3]. Continuing to boost the natural accuracy of certifiably robust models is an ongoing research effort in the community. As of now, the natural accuracies might seem low, but these have been consistently improving and laying the foundations for further improvements. As such, we hope that our results are seen in the light of the norms and current state-of-art in the certifiable robustness community.


[1] Fu et al. DreamSim: Learning New Dimensions of Human Visual Similarity using Synthetic Data, NeurIPS 2023 Spotlight.
[2] Hu et al. Unlocking Deterministic Robustness Certification on ImageNet. NeuIPS 2023.
[3] Chen et al. Symbolic Discovery of Optimization Algorithms. NeuIPS 2023.
[4] Apple. CSAM Detection - Technical Summary. https://www.apple.com/child-safety/pdf/CSAM_Detection_Technical_Summary.pdf 2021.
[5] Struppek et al. "Learning to break deep perceptual hashing: The use case neuralhash." ACM Conference on Fairness, Accountability, and Transparency. 2022.

---

> ### Author Response · Authors · 2023-11-17
> **Summary of changes in the paper**
>
> - We added a reference for the explanation and instances of the 2AFC dataset to the introduction.
> - To evaluate LipSim on more attacks we added the Momentum Iterative Attack (MIA).
> - We also added the certified robustness results of LipSim on another 2AFC dataset (called BAPPS).
> - We added the theoretical comparison of LipSim with two previous works on the certified robustness of perceptual metrics which have looser bounds.
> - We added an empirical comparison between LipSim and a recent empirical defense (R-LPIPS).
> - We added KNN (k-nearest neighbors algorithm) as a new application for LipSim besides Image Retrieval and compared our performance with DreamSim.
> - We changed the color of the bar chart in Figure 3.a.

---

### Author Response · Authors · 2023-11-21

Dear Reviewers,
Thank you once again for your comments and feedback. We would like to kindly remind you that we are approaching the end of the rebuttal period, so please don’t hesitate to reach out if you still have any questions or comments about our work. We are committed to addressing all concerns related to our work.

---

### Author Response · Authors · 2023-11-22

As the rebuttal period concludes, we believe that our responses have fully addressed each concern raised by the reviewers. Below is a summary of our rebuttal:
- We provided an explanation and a real-world example (Apple NeuralHash) to further demonstrate the importance of a robust perceptual metric.
- We provided information about the trade-off between natural performance and robustness (Lipsim vs. Dreamsim is 85.09% / 96.16% for natural accuracy and 65.62% / 0.93% for robustness). In addition, LipSim provides a certified score of 67.32% while DreamSim does not have any theoretical guarantees.
- In addition to using AutoAttack, which is an ensemble of four attacks, we added _two more attacks_ to evaluate the empirical robustness of LipSim.
- We provided certified robustness by leveraging LipSim on another available 2AFC dataset (BAPPS).
- We compared LipSim with previously proposed certified metrics and showed that bounds proposed by previous metrics are ten times looser and they are computationally very heavy. LipSim is the first robust perceptual metric with certificates and non-trivial bounds.
- We compared LipSim with previous non-certified metrics (R-LPIPS) in terms of empirical robustness on both the BAPPS and NIGHT datasets.
- We provided results on zero-shot classification as another application for LipSim besides image retrieval.
- We included all these new results in the revised paper.

---

### Meta-Review · Area_Chair_bSRv · 2023-12-10

**Metareview:**

The paper proposes a provably Lipschitz continuous image similarity measure that is intended to improve DreamSim in terms of robustness under attack. The idea is, simply speaking, to use a Lipschitz network (proposed by Araujo et al. (2023)) to distill DreamSim and then finetune on the NIGHTS daaset. The idea behind the paper is a simple combination of prior works, yet summarizes the mathematical insights to yield a robust similarity measure. The evaluation is done using AutoAttack which is the SotA attack for image classification.
The reviewers provide borderline scores with several suggestions for additional experiments. In the rebuttal, the results to these experiments are provided, showing, on more data, the improved robustness of the proposed measure over DreamSim while DreamSim provides a better alignment with human perception on clean data. After reading the paper and additional results thoroughly, my impression is that the experimental validation is actually very solid, including not only l_2 but also l_infinity bounded attacks. The only question I have after the rebuttal is concerning the attack budget: the improved robustness is mostly shown on attacks with a large attack budget, while for clean images DreamSim is better. What is the attack budget from which on LipSim provides more reliable results?
Although the method is a straight-forward application of Lipschitz networks, it provides complementary information to DreamSim for model evaluation and can therefore be a valuable tool in practice.

**Justification For Why Not Higher Score:**

The submission proposes a nice adaptation of DreamSim that is certifiably robust. Yet, the proposed adaptation is rather simple and robust image similarity metrics are of interest to a limited part of the community.

**Justification For Why Not Lower Score:**

The proposed method is theoretically justified and empirically validated. The paper is well written and the reviewer's questions have been answered.

---

### Decision · Program_Chairs · 2024-01-16

Accept (poster)